# Antisense Oligonucleotide-Based Therapy of Viral Infections

**DOI:** 10.3390/pharmaceutics13122015

**Published:** 2021-11-26

**Authors:** Woan-Yuh Tarn, Yun Cheng, Shih-Han Ko, Li-Min Huang

**Affiliations:** 1Institute of Biomedical Sciences, Academia Sinica, 128 Academy Road, Section 2, Nankang, Taipei 11529, Taiwan; 2Department of Pediatrics, National Taiwan University Children’s Hospital, National Taiwan University College of Medicine, 8 Chung-Shan South Road, Taipei 10002, Taiwan; cyskynet@gmail.com; 3Biomedical Translation Research Center, Academia Sinica, Taipei 11529, Taiwan; shko@gate.sinica.edu.tw

**Keywords:** virus, virus-host interaction, RNA therapeutics, antisense oligonucleotide, drug delivery

## Abstract

Nucleic acid-based therapeutics have demonstrated their efficacy in the treatment of various diseases and vaccine development. Antisense oligonucleotide (ASO) technology exploits a single-strand short oligonucleotide to either cause target RNA degradation or sterically block the binding of cellular factors or machineries to the target RNA. Chemical modification or bioconjugation of ASOs can enhance both its pharmacokinetic and pharmacodynamic performance, and it enables customization for a specific clinical purpose. ASO-based therapies have been used for treatment of genetic disorders, cancer and viral infections. In particular, ASOs can be rapidly developed for newly emerging virus and their reemerging variants. This review discusses ASO modifications and delivery options as well as the design of antiviral ASOs. A better understanding of the viral life cycle and virus-host interactions as well as advances in oligonucleotide technology will benefit the development of ASO-based antiviral therapies.

## 1. Introduction

In 2019, the World Health Organization listed eight RNA viruses, including influenza, Zika, dengue, and severe acute respiratory syndrome-coronavirus (SARS-CoV), as the top threats to global health [1]. After that, coronavirus disease 2019 (COVID-19) has caused a global pandemic. Thus, it is critical for us to achieve more rapid development of vaccines and antivirals against newly emerging viruses. Vaccination is the most effective way to protect against infectious diseases, and conventional vaccines have been used to prevent infection of numerous viruses from polio to flu, although their effectiveness varies. Recently, RNA-based vaccines have demonstrated their efficacy against the outbreak of SARS-CoV-2 [2]. Nonetheless, antiviral medicines, including small-molecule drugs and biologics, are still needed to combat emerging viral pathogens or prevent disease progression.

Small-molecule drugs are the main class of therapeutics for treating various diseases and conditions. These drugs target and bind allosterically to disease-associated proteins and receptors or inhibit the activities of metabolic enzymes. As of 2019, approximately 90 antiviral drugs had been approved worldwide for the treatment of humans [3]. Antiviral drugs are designed to block viral entry or inhibit virus-encoded enzymes that are required for viral propagation. However, the majority of antiviral drugs have been developed for chronic infections caused by human immunodeficiency virus (HIV), hepatitis B and C viruses (HBV and HCV), and herpesviruses, yet only a few have been developed to treat acute infections such as influenza. The fact that viruses evolve rapidly and encode only a few enzymes that can be targeted has presented challenges for the development of antivirals. Currently, integration of knowledge across multiple disciplines, high-throughput screening, and artificial intelligence-assisted drug design has facilitated the development of small-molecule antiviral drugs [4]. Moreover, biologics-based drugs such as peptides, cytokines, monoclonal antibodies, and nucleic acids have become the fastest-growing class of therapeutic agents. For example, peptides derived from viral capsid proteins or monoclonal antibodies against viral surface proteins have been used to block access to host-cell receptors [5].

Nucleic acid-based therapeutics can target a genetic culprit via complementary base-pairing, and in this regard, this approach is superior to small-molecule or protein drugs—particularly for antiviral treatment [6,7]. Nucleic-acid therapeutics exploits various types of DNA or RNA molecules, including antisense DNA oligonucleotides (ASOs), short interfering RNA (siRNA), microRNAs (miRNAs), single-guide RNAs (sgRNAs), ribozymes, aptamers, and triplex-forming oligonucleotides (Table 1). The most widely used nucleic acids are ASOs and siRNAs that cause target RNA cleavage/degradation or block mRNA processing or translation. Antagomirs are a class of ASOs that particularly silence cellular microRNAs. This review focuses on ASOs that are used for antiviral strategies. Ribozymes are catalytic RNAs that can cleave target RNAs. A clinical trial was conducted to test the efficacy of a hammerhead ribozyme targeting the mRNA encoding the HIV co-receptor CCR5 in combination with other therapeutic nucleic acids in 2010 [8]. An aptamer is a single-stranded DNA or RNA that folds into a specific structure that allows high-affinity binding to a protein or other biomolecule [9]. For example, a 33-nucleotide pseudoknot RNA aptamer can bind the HIV-1 reverse transcriptase and thereby inhibit the release of viral particles [10]. In addition, an in vitro transcribed mRNA encoding a viral envelope protein can be encapsulated into lipid nanoparticles (LNPs) for subsequent use in a vaccine [11]. Analogously, an in vitro transcribed mRNA encoding the programmable nuclease Cas9 together with a single-guide RNA can be used for genome editing [12]. It is conceivable that RNA technology would show its power in future therapeutics.

## 2. Development and Delivery of Therapeutic ASOs

ASO is a single-stranded DNA of 12–25 nucleotides in length targeting diseases-associated transcripts [13,14,15,16]. The first ASO that inhibits the replication of Rous sarcoma virus and cell transformation was reported in 1978 [17]. ASOs modulate target gene expression via one of the following mechanisms [13,14,15,16]. (1) ASO forms a hybrid with the target RNA, inducing RNA cleavage by RNase H, which is an endonuclease cleaving the RNA strand of a DNA-RNA duplex. (2) ASO base-pairs with a functional *cis*-element of the target RNA and hence blocks access of cellular machinery to the RNA. Such steric-blocking ASOs may cause splice isoform switch or translation inhibition or may alter the stability of the target RNA (Figure 1).

### 2.1. Modifications of ASOs

Chemical modifications have been developed to improve the pharmacokinetic properties of ASOs, including stability, specificity, and membrane permeability, and minimize their cytotoxicity. Three generations of ASOs have been broadly classified with respect to the types of modifications [18,19,20]. In the first-generation ASOs, one of the non-bridging oxygen atoms in the phosphodiester bond are replaced, resulting in a phosphoramidate, methylphosphonate, or phosphorothioate (PS) linkage (Figure 2A, PS). The PS linkage improves membrane penetration of ASOs and does not interfere with RNase H-mediated RNA cleavage [18,19,20]. PS-ASOs can bind proteins in plasma, thereby preventing rapid clearance of the ASOs from the circulatory system [21,22]. PS chirality (*R*p and *S*p) may influence the pharmacological properties of ASOs, such as the stability and RNase H1 cleavage patterns [23,24]. The second-generation ASOs are modified with an alkyl moiety, such as a methyl (2′-OMe) or methoxyethyl (2′-MOE) group at the 2′ position of the ribose (Figure 2A). These ASOs have greater affinity for their target RNAs and lesser cytotoxicity but cannot recruit RNase H1 for RNA cleavage [25,26]. The third-generation ASOs include locked nucleic acid (LNA), peptide nucleic acid (PNA), and phosphorodiamidate morpholino oligomer (PMO) (Figure 2A). LNAs contain a constrained ribose ring having a O2′-C4′-methylene linkage. PNAs have a peptide-like N-(2-aminoethyl)glycine linkage to replace the ribose-phosphate DNA backbone. PMOs contain a backbone of morpholine rings connected by phosphorodiamidate linkages. These modified ASOs exhibit high affinity to targets, improved pharmacokinetic profiles and nuclease resistance compared with conventional ASOs and essentially cause steric hindrance instead of RNase H-mediated cleavage [18,19,20]. However, uncharged nucleic acids, such as PNAs and PMOs, exhibit poor cellular uptake and need a conjugate or carrier (see below). Nevertheless, piperazine groups have been introduced along the backbone of PMOs to provide a net positive charge [27] (Figure 2A). Furthermore, a combination of the structural elements of 2′MOE and LNA yields highly nuclease-resistant constrained nucleic acids, such as 2′-4′-constrained 2′O-ethyl nucleotide (cEt) [28] (Figure 2A).

To improve potency and efficacy, chimeric ASOs have been developed having different modifications in the base, phosphodiester linkage, and deoxyribose moiety. Gapmers have been designed to consist of a central short region of deoxyribonucleotides flanked by a stretch of ribonucleotides in which the ribose ring is modified with 2′-OMe, 2′-MOE or LNA [18,19,20,29]. Therefore, a gapmer can induce RNase H-mediated cleavage of the target RNA with a relatively greater binding affinity and specificity than conventional ASOs (Figure 1). Using a cell-based assay, a gapmer targeting the internal ribosome entry site of HCV exhibited potent antiviral activity (50% effective concentration, 4 nM) [30]. Mipomersen, which targets apolipoprotein B-100 mRNA, is an FDA-approved drug to treat familial hypercholesterolemia; it is a gapmer consisting of a 5′methyl (m^5^)-C/U-containing PS-ASO “gap” and 2′-MOE nucleotides at both ends [31,32]. Many ASOs act as a steric blocker (Figure 1). For example, the FDA-approved drug nusinersen is a m^5^C/2′-MOE/PS-ASO for the treatment of spinal muscular atrophy; this ASO corrects the aberrant splicing of *SMN2* by preventing the binding of splicing suppressors to its intronic silencer [33,34]. Recently, PS-ASO gapmers carrying 2′-deoxy-2′-fluoroarabinonucleotides (FANA; Figure 2A) at both ends were developed to target the HIV-1 genome; they could both activate RNase H-mediated RNA cleavage and sterically block the dimerization of viral genomic RNA during virion assembly [35]. Additional examples for antiviral ASOs are described in Section 3.

### 2.2. Bioconjugations of ASOs

Nucleic acid-based drugs generally enter cells via endocytosis; most of these therapeutics must be formulated as a bioconjugate to facilitate receptor-mediated endocytosis and/or increase their lipophilicity [36,37,38] (Figure 2B). For example, conjugation of an siRNA with anandamide, folate, or cholesterol enables efficient uptake of the siRNA by cells, likely via the corresponding conjugate-specific receptor [39]. Notably, a PS backbone assists the membrane translocation of ASOs; thus, bioconjugation and/or delivery agents are often dispensable for PS-ASOs [21,22]. Therapeutic oligonucleotides also can be conjugated to certain ligands that bind cell type-specific receptors. For example, glycoproteins terminating with *N*-acetylgalactosamine (GalNAc) can be recognized by the asialoglycoprotein receptor (ASGPR), which exists primarily on the cell-surface of hepatocytes [40]. Therefore, conjugation with GalNAc can promote the uptake of siRNAs and ASOs by hepatocytes via ASGPR-mediated endocytosis. GalNAc-2′-MOE-ASOs targeting the mRNA encoding the liver glucagon receptor have been designed for treatment of type 2 diabetes [41]. CpG dinucleotides can lead to the uptake of conjugated oligonucleotides by dendritic cells or macrophages that express innate immune receptors [42]. Besides the aforementioned conjugates, a number of cell-penetrating peptides (CPPs) have been developed to enhance drug delivery, including polycationic HIV-1 Tat peptide, the hydrophobic residue-containing peptide penetratin that is derived from the *Drosophila* antennapedia homeodomain, and artificial poly-arginine peptides [43]. CPPs may undergo endocytosis or directly penetrate cells [43]. Composite CPPs consisting of penetratin and 6-aminohexanoic-spaced oligo-arginine (RXR) have been used to enhance the efficiency of delivery of charge-neutral PMOs or PNAs to cells in vivo or in culture [44]. Various conditionally activatable CPPs have been designed for selective delivery. For example, a pH-sensitive transportan CPP bearing lysine-to-histidine substitutions can enter cells under acidic conditions, such as the tumor microenvironment [45]. Finally, because guanidinium groups of arginine-rich peptides are critical for peptide translocation across the plasma membrane, a synthetic octa-guanidine dendrimer has been conjugated to PMOs, and such conjugates are called vivo-PMOs [46]. Vivo-PMOs, although widely used for transient gene silencing in vitro, cause coagulation owing to dendrimer clustering in animals; supplementation of Vivo-PMO with anticoagulants may counteract its toxicity [47].

### 2.3. Vehicle-Mediated Delivery of ASOs

Oligonucleotide bioconjugates offer the potential for enhanced drug delivery, but recent advances in nanotechnology have further benefited the transport of therapeutic ASOs across biological barriers and improved their pharmacokinetics in circulating blood. Several nanoparticle-mediated delivery systems have been developed [15,18,36,37,38] (Figure 2C).

Cationic polymer nanocarriers are formed via ionic interactions between negatively charged ASOs and positively charged macromolecules such as polyethylenimine (PEI). PEI promotes cellular delivery of ASOs but is somewhat cytotoxic. Modification with phospholipid (such as dioleoylphosphatidylethanolamine, DOPE) or copolymerization with polyethylene glycol (PEG) can enhance the efficiency of PEI in ASO delivery and reduce its cytotoxicity [48]. Moreover, bioconjugation with cell-binding ligands such as transferrin, antibodies, or carbohydrates can facilitate receptor-mediated uptake of nanocarriers [18].

LNPs are nanoparticles mainly constructed with lipids. Among them, liposomes are spherical vesicles comprising single or multiple lipid bilayers. ASOs can be carried in the aqueous space encapsulated by artificial liposomes. Most liposomes formulated for RNA delivery comprise both cationic lipids and neutral lipids such as DOPE; such a lipid combination enhances the transfection efficiency and reduces the cytotoxicity of liposomes [49]. At present, LNPs represent a highly potent RNA delivery vehicle; for such LNPs, nucleic acids are organized in inverse lipid micelles inside the nanoparticle [50]. LNPs can be made of ionizable lipids, cholesterol, phospholipids, and PEG-lipid conjugates [51]. Ionizable lipids are pH-sensitive, and they enable endosomal escape after cellular uptake [52]. As noted above, conjugation of a ligand to PEGylated liposomes promotes cell-specific targeting.

Exosomes are naturally secreted extracellular vesicles that transfer macromolecules between cells [53]. For drug delivery, exosomes have both advantages and drawbacks. For example, they have inherent anti-inflammatory properties, can traverse biological membranes such as the blood-brain barrier, and can be produced in an autologous manner, but they are heterogenous and uneasy for large-scale production [54]. Exosomes can be modified through chemical methods or genetic engineering. Fusion of green fluorescent protein to the exosome surface protein CD63 allows tracking of the exosome and monitoring of cargo delivery [55]. Coating of exosomes with cationic lipids and a pH-sensitive amphipathic peptide can enhance cellular uptake and fusion with endosomes and subsequent cargo release [56]. Moreover, exosomes can target specific cell types upon certain modifications. For example, exosomes carrying a peptide derived from a rabies virus glycoprotein have been used to deliver siRNAs that target the Alzheimer’s disease-associated BACE1 in neurons [57].

## 3. ASOs Targeting Viruses

The development of more efficacious treatments against various viral diseases from acute to persistent infection is still in high demand. Among different nucleic acid-based therapies, ASOs directly act on viral genomic RNA or transcripts (Figure 3) and can be rationally designed for any new virus (or variants) or a reemergent virus. In 1998, vitravene (fomiversen), the first FDA approved ASO, was developed for the treatment of cytomegalovirus retinitis in immunocompromised HIV patients [58]. Examples given below describe many additional ASOs and related therapeutic nucleic acids that have been designed for antiviral treatment.

### 3.1. Coronaviruses

Coronaviruses can cause a range of illnesses, ranging from the common cold to severe diseases such as SARS and COVID-19. Coronaviruses have a positive-strand RNA genome of ~30 kb. The genome contains two large overlapping open reading frames (ORFs), namely ORF1a and ORF1b, at the 5′ terminus, followed by ORFs encoding structural proteins and accessory factors. The two large ORFs generate two polyproteins, pp1a and pp1ab; the latter results from a programmed -1 ribosomal frameshift at the short overlap of ORF1a and ORF1b [59]. The two polyproteins undergo proteolytic cleavage to generate 16 non-structural proteins, and pp1ab is responsible for the formation of the replicase machinery [60]. Moreover, to produce structural and accessory proteins, each coronavirus must generate a nested set of positive-strand subgenomic mRNAs. Synthesis of the templates of these mRNAs involves discontinuous transcription, for which multiple transcription regulatory sequences in the genomic RNA play an important role by base-pairing with nascent subgenomic RNAs.

An early study revealed that arginine-rich CPP-conjugated PMOs targeting the translation start site of the murine coronavirus replicase polyprotein could reduce viral titer in cultured cells [61]. Similar PMOs have been designed to target a region containing the transcription regulatory sequence of SARS-CoV, and they can potently decrease viral amplification [62]. A CPP-conjugated PNA targeting the ribosome frameshifting region could suppress SARS-CoV replication [63]. The COVID-19 pandemic has prompted the rapid development of therapeutic strategies against SARS-CoV-2. A combination of cryo-electron microcopy and molecular modeling has revealed the tertiary structure of the frameshift stimulation element of SARS-CoV-2. LNA-modified ASOs targeting the structure of this element can disrupt translational frameshifting and hence inhibit viral replication [64]. Another report showed that a 2′-OMe/SP-ASO conjugated with four 2′-5′-oligoadenylates that can induce RNase L-mediated cleavage and degradation of the SARS-CoV-2 envelop and spike RNAs can effectively inhibit viral propagation in pseudovirus infection models [65].

### 3.2. Dengue Virus

Dengue infection occurs in tropical and subtropical areas and causes fever and flu-like symptoms. Dengue viruses have a 10.7 kb positive-strand RNA genome encoding three structural proteins and seven nonstructural proteins. The 5′ and 3′ untranslated regions, respectively, fold into conserved structures that are essential for viral viability. The 5′-most stem-loop acts as a promoter of viral RNA replication. RNA replication involves genome cyclization, which is mediated by the interaction between the complementary 5′ and 3′ cyclization sequences [66]. Arginine-rich CPP-PMOs that respectively target the 5′ or 3′-terminal stem-loop or 3′ cyclization sequence of the Dengue genome can inhibit viral replication and decrease viral titer in cultured cells [67,68]. Analogously, a recent study showed that 3′ stem-loop-targeting vivo-PMOs can potently inhibit Dengue replication in dendritic cells that are primary target cells of dengue infection [69].

### 3.3. Respiratory Syncytial Virus (RSV)

RSV causes lower respiratory tract disease that most often affects children and older individuals. RSV has a negative-strand RNA genome of ~15 kb. After entry into the host cell, the viral nucleocapsid and polymerase are delivered into the cytoplasm. Viral RNA-dependent RNA polymerase (RdRp) transcribes the viral genome into mRNAs that encode viral proteins and synthesizes the antigenome, which serves as the template for genome synthesis [70]. ASOs that induce RNase H-mediated cleavage/degradation of RSV genomic RNA can inhibit RSV replication [71]. A CPP-conjugated PS-PMO could inhibit RSV replication in mice by suppressing the translation of RSV L mRNA [72]. The intranasal route is a rational choice for delivery of antiviral drugs against respiratory infections. One study has demonstrated that intranasal administration of siRNAs that knock down RSV phosphoprotein expression can effectively reduce RSV infection and prevent pulmonary pathology in mice [73].

### 3.4. Influenza

Influenza viruses cause a contagious respiratory illness ranging from mild to severe. The influenza genome comprises eight negative-strand RNA segments, each of which encodes one or two proteins. In contrast to most RNA viruses, the influenza RNAs are transcribed and replicated by viral RdRp in the nucleus. For viral synthesis, RdRp uses a cap-snatching mechanism to prime transcription [74]. Notably, all eight viral RNAs contain conserved sequences respectively at their 5’ and 3’ termini [74]. Although neuraminidase inhibitors are the most frequently used anti-influenza drugs, other antiviral strategies are still necessary [75]. CPP-conjugated PMOs targeting the 3′ conserved region of the nucleocapsid mRNA can reduce the viral titer [76]. Using titanium dioxide (TiO_2_) as a nanocarrier, polylysine-linked ASOs targeting the same conserved region exhibited potent antiviral activity with little cytotoxicity [77]. Furthermore, conjugation of ASOs with a peptide binding to the influenza hemagglutinin can increase the efficiency of ASO delivery in mice [78]. Radavirsen is a positively charged and CpG-containing PMO that blocks the translation of the M1/M2 matrix proteins and can synergize the effect of neuraminidase inhibitors in influenza-infected animal models [79], showing its clinical efficacy.

### 3.5. Ebola Virus

Ebola virus is a rare but deadly virus that causes coagulation abnormalities, leading to hemorrhagic fever. A recent outbreak occurred in West Africa from 2014 to 2016. Ebola and its relative of the Marburg virus belong to the *Filoviridae* family, and these viruses have a negative-strand RNA genome of 19 kb encoding seven proteins. Among them, VP24 and VP35 antagonize the innate antiviral immune response via multiple pathways and are responsible for the extreme virulence of Ebola virus [80]. Essentially, VP24 inhibits the activation of interferon-stimulated genes by preventing nuclear import of a key transcription factor STAT1, whereas VP35 interacts with double-stranded RNA ends to prevent sensing by cellular pattern recognition receptors such as retinoic acid inducible gene-I (RIG-I) [81,82]. Positively charged PMOs targeting VP35 mRNA could protect mice from infection-induced lethality [83]. Moreover, targeting both VP24 and VP35 achieved postexposure efficacy against Ebola virus in nonhuman primates, indicating positively charged PMOs as effective therapeutic agents [84].

### 3.6. HBV

HBV, a prototype virus of the *Hepadnaviridae* family, has a 3.2 kb partially double-stranded, relaxed circular DNA genome [85]. After infection, the viral genome is converted to covalently closed circular DNA in the nucleus, and this DNA serves as the template for synthesis of pregenomic RNA and subgenomic viral transcripts. Viral replication occurs by reverse transcription of pregenomic RNA. Chronic HBV infection may lead to liver failure and liver cancer. Interferon and nucleoside/nucleotide analogs are the most commonly used therapeutics [86]. Developing new therapeutic strategies is still necessary, however, owing to drug resistance. In a pioneering study, a PS-ASO complementary to the HBV polyadenylation signal sequence complexed to a ASGPR ligand and polylysine reduced viral surface antigen (HBsAg) expression and blocked HBV replication in cultured cells [87]. Recently, GalNAc-conjugated LNA-ASOs that can destroy viral mRNAs via RNase-H-mediated degradation showed a significant HBsAg reduction in HBV-infected mice [88]. Notably, the uncapped 5′-triphosphate of RNA can activate the RIG-I-dependent antiviral type-I interferon response [89], and a recent study demonstrated that 5′-triphosphate-modified siRNAs can target HBV and meanwhile elicit an antiviral immune response [90]. More recently, the phase II clinical trial shows that the ASO bepirovirsen targeting a conserved sequence present in all HBV mRNAs effectively suppresses HBV replication in chronically infected patients [91].

### 3.7. HIV

HIV causes life-threatening immunodeficiency syndrome and thus has long attracted innovative development of antiviral drugs. HIV is a lentivirus that infects and subsequently depletes CD4^+^ helper T cells. Upon entry into these T cells, the viral RNA genome is reverse transcribed into DNA by viral reverse transcriptase. The resulting viral double-stranded DNA is integrated into the host genome by the viral integrase and host factors [92]. Many anti-HIV drugs have been developed to target viral enzymes (such as reverse transcriptase, integrase and protease) or prevent viral entry by blocking the T-cell receptors CD4 or CCR5 [93]. Numerous nucleic acid-based drugs have also been designed for HIV treatment. For example, early studies showed that an siRNA targeting the HIV genome or HIV Gag-p24 could inhibit viral replication [94,95]. Further, bimodular aptamers containing a 5′ stem-loop connected to a 3′-guanosine quadruplex were developed to inhibit the reverse transcriptase of diverse primate lentiviruses [96]. Similarly, a guanine-tethered ASO that could form an DNA-RNA quadruplex structure with the HIV RNA genome could inhibit reverse transcription in cis [97]. Finally, recent studies continue to demonstrate the therapeutic potential of new-generation ASOs such as FANA/PS-ASOs [35] (see Section 2.1).

## 4. ASOs Targeting Host Factors

Viruses exploit host-cell organelles and molecular machineries to complete their life cycle and transmission, and viruses also modulate cellular signaling pathways for their own benefit. Therefore, ASO-mediated transient knockdown of relevant cellular factors may prevent viral propagation (Figure 4). Certain cellular factors are commonly hijacked by different viruses and thus could be used for developing broad-spectrum antiviral agents. Examples given below include gapmers, antagomirs, and splicing-switching ASOs that target the transcripts encoding those cellular factors (e.g., cell-surface receptors, transport systems, signaling factors) or miRNAs.

### 4.1. Niemann-Pick C1 (NPC1)

NPC1 is a multi-transmembrane protein essential for cholesterol transport from late endosomes and lysosomes and regulates cellular lipid homeostasis [98]. NPC1 mutations cause accumulation of cholesterol and other lipids in various tissues. The lysosomal accumulation of lipids in Niemann-Pick type C disease with NPC1 mutations leads to neurological impairments and progressive neurodegeneration [99]. Moreover, NPC1 and NPC1-like protein participate in the infection by various viruses. NPC1 serves as a fusion receptor for filovirus. Ebola virus is internalized via a micropinocytosis-like process and is subsequently transported to late endosomes [100]. The Ebola virus glycoprotein (GP) is required for virion/cellular membrane fusion. Upon proteolytic processing of the GP1 subunit, its receptor-binding domain interacts with endosomal NPC1 for viral entry and subsequent release of the viral nucleoprotein in the cytoplasm followed by replication of viral genomic RNA [100]. Fibroblasts derived from patients with Niemann-Pick type C disease are resistant to filovirus infection [101]. LNA-PS-modified ASOs targeting NPC1 mRNA can interfere with the cellular entry of a filovirus glycoprotein-pseudotyped virus [102]. Similar to Ebola virus, SARS-CoV family viruses also utilize the endocytic machinery to enter host cells. However, cellular entry of SARS-CoV does not require NPC1 per se but rather relies on increased cathepsin L activity in NPC1-containing late endosomes or lysosomes [103]. Accordingly, recent studies demonstrated that an NPC1 inhibitor could suppress the cellular entry of pseudotyped SARS-CoV-2 [104,105]. Therefore, antisense strategies targeting NPC1 may treat and prevent human coronavirus infections.

### 4.2. Raf-1

The serine/threonine kinase Raf1 is a downstream effector of RAS in the mitogen-activated protein kinase pathway. Ras/Raf/MEK/ERK signaling regulates numerous cellular processes such as proliferation and differentiation in response to extracellular stimuli [106]. A number of viruses exploit this pathway to modulate their infectious cycle. HCV activates Raf-1 by its core protein and thereby regulates hepatocyte growth and differentiation [107]. Conversely, Raf-1 participates in HCV replication via its interaction with the viral nonstructural protein 5A in the replication complex [108]. Thus, inhibition of Raf-1 attenuates viral replication. Notably, activation of the Ras/Raf/MEK pathway downregulates the expression of interferon-stimulated genes that are critical for the innate immune response. Therefore, HCV infection attenuates interferon signaling by activating Raf-1 and hence benefits viral propagation [109]. Via a similar mechanism, vesicular stomatitis virus exhibits tropism for malignant cells in which Raf-1 is activated [110]. In addition, influenza virus activates the Raf/MEK/ERK cascade for nuclear export of its ribonucleoproteins, which is required for viral propagation [111]. Raf-1 also contributes to infection by Japanese encephalitis virus, and a Raf-1-targeting ASO could reduce viral propagation and inhibit virus spread from the periphery to the brain in a mouse model of infection with JEV [112]. This result thus implies that Raf-1 is a potential target for broad-spectrum antiviral agents.

### 4.3. The Heat-Shock Protein MRJ

Heat-shock proteins function as molecular chaperones to maintain proteostasis [113]. A number of viruses modulate the cellular heat-shock response or take advantage of cellular heat-shock proteins to overcome host environmental challenges and complete their life cycle [114,115]. An early study revealed that the human heat-shock protein DNAJB6 (also termed MRJ) is critical for nuclear import of the HIV-2 preintegration complex via its interaction with viral Vpx protein [116]. Notably, MRJ has two splice isoforms that exert different effects on viral infection [115]. Isoform switching from the C-terminally truncated MRJ-S to full-length MRJ-L occurs during monocyte differentiation into macrophages, which are target cells for HIV [117]. Accordingly, individuals with a higher level of MRJ-L in macrophages are more susceptible to HIV infection [118]. MRJ-L possibly facilitates the nuclear import of the HIV preintegration complex via its C-terminal nuclear localization signal [118]. Similarly, it promotes nuclear entry of the cytomegalovirus primase [119]. MRJ-L is also essential for subgenomic mRNA production and viral propagation of RSV even though the RSV life cycle is completed in the cytoplasm [117]. Conversely, the MRJ-S isoform is involved in Dengue RNA replication and virion production [120]. Therefore, manipulating MRJ isoform expression is a potential antiviral strategy. A splicing-switching vivo-PMO that suppresses MRJ-L expression could reduce the propagation of pseudotyped HIV-1 in THP-1 monocytes and RSV in Hep2 epithelial cells [117], indicating its potential efficacy as a broad-spectrum antiviral agent.

### 4.4. miR-122

miRNAs regulate gene expression at the post-transcriptional level via binding to the 3′ untranslated region of target mRNAs. Therefore, miRNAs can regulate the pathogenesis of a broad range of viruses; most of them downregulate viral translation and replication [121]. However, viruses can modulate the expression of host miRNAs, most of which are involved in antiviral innate immunity. Notably, HCV infection upregulates miR-122, which is abundant in hepatocytes and regulates liver homeostasis [122]. miR-122 in turn binds to sites upstream of the internal ribosome entry site in the 5′ non-coding region of the viral RNA genome and hence increases viral RNA stability and upregulates viral RNA translation and replication [123,124]. Host factors that are involved in miRNA biogenesis, such as Dicer, TRBP and Ago2, also contribute to viral RNA accumulation [125]. An early study demonstrated that miR-122-targeting ASOs downregulate HCV replication in liver [123]. Subsequently, various modifications were introduced into miR-122 ASOs, including 2′-MOE, LNA and PNA [126,127]. Miravirsen, a 15-nucleotide LNA/PS-modified ASO targeting miR-122, has shown potential for treating chronic HCV in clinical trials [128]. Recently, bioconjugation of miR-122 antagomirs with GalNAc has demonstrated their hepatic specificity and improved pharmacokinetics [129]. HCV also upregulates miR-146a in hepatocytes; miR-146a may subsequently interfere with the immune response, promote viral assembly, and deregulate liver metabolism [130]. Therefore, miR-146a may also be a potential target for anti-HCV therapeutics. Antagomirs may be used alone or in combination with other antiviral agents for treatment of chronic HCV.

### 4.5. Other Host Factors Targeted by ASOs

ASGPR is a candidate receptor for HBV, and its major subunit ASGPR1 is upregulated in cells of HBV-infected patients. ASOs targeting ASGPR1 mRNA in HBV-infected human hepatocellular carcinoma cells can reduce the level of viral antigens and DNA [131]. A subcellular proteomic screen implicated the involvement of programmed cell death 5 (PDCD5) during influenza virus infection. PDCD5 may suppress tumors and pathogenic T cells by inhibiting cell proliferation and inducing apoptosis. Knockdown of PDCD5 mitigated influenza HIN1 propagation in cultured cells [132]. A subsequent study showed that PDCD5-targeting ASOs can downregulate PDCD5 in lung tissue and hence protect mice from influenza virus infection [133].

## 5. Conclusions and Perspectives

Rapid and effective treatment against emerging infectious diseases is predicted to be critical for public health worldwide. Development of new antivirals is also important for fighting chronic infections, seasonal respiratory viruses, and drug-resistant strains. Nucleic-acid antivirals may fulfill this need because they can be rationally designed based on viral genome sequences. Successful development of such antivirals will require a comprehensive understanding of viral biology and advances in both nucleic-acid chemistry and nanotechnology. ASOs represent a type of drugs for personalized and precision medicine, and they can be designed and manufactured quickly. However, a number of disadvantages or concerns still remain. ASOs are susceptible to degradation in plasma and may have off-target effects or activate the immune system [19]. Moreover, ASOs have a high propensity to accumulate in certain organs such as liver and kidney and hence cause hepatotoxicity or renal dysfunction [134]. Nevertheless, a recent study shows that mesylphosphoramidate linkages increase plasma stability and reduce pro-inflammatory effects and cytotoxicity [135]. Therefore, advances in chemical modifications and delivery carriers are expected to improve the potency and safety of ASOs [136].

Artificial intelligence techniques and cutting-edge RNA sequencing methodologies have already provided insights into the molecular features of viral life cycles and the heterogeneity in virus-cell interactions. These findings may inform the identification of new targets for antiviral therapeutics. Transcriptional profiling at the single-cell level enables the analysis of cell diversity and heterogeneity and reveals detailed molecular mechanisms underlying viral infection. For example, single-cell RNA-seq has revealed cell type-specific differences between influenza-infected and bystander lung cells, which may benefit the future design of targeted therapies [137]. Moreover, single-molecule RNA-seq has revealed the complex transcriptome of SARS-CoV-2 that is generated by discontinuous transcription and identified previously unknown ORFs [138]. RNA sequencing-based structural mapping has revealed structural landscapes of the SARS-CoV-2 genome. Subsequently, a deep-learning strategy identified a set of RNA-binding proteins that interact with those structural elements. ASOs targeting newly identified viral ORFs or host factors may further inform the development of antiviral therapeutics [139].

The development of new oligonucleotide chemical modifications and nanomaterials will continue to benefit future applications of nucleic acids as antivirals. A new generation of nucleoside analogs and novel internucleoside linkages have led to improved target affinity, cellular delivery, and pharmacokinetics. For example, a high-affinity STAT3-targeting gapmer consisting of a PS-DNA “gap” and flanking 2′-4′-constrained-2′O-ethyl nucleotides can be delivered into cells without a carrier and has been the subject of a clinical trial for treatment of lymphoma and lung cancer [140]. HIV-targeting, FANA-modified ASOs can also be internalized by cells without a carrier OR carriers [35]. Inhalable or intranasal delivery of antiviral ASOs may be used for pulmonary delivery of ASOs targeting viral respiratory infections [141]. Finally, LNPs are one of the most promising carriers for ASO delivery and can be modified with cell-specific ligands or antibodies for selective ASO delivery to infected tissues—for example, GalNAc for liver cells and anti-CD4 for CD4^+^ T cells [142]. Growing knowledge and emerging technologies would benefit clinical translation of ASOs and other RNA-based therapies in the future.

## Figures and Tables

**Figure 1 pharmaceutics-13-02015-f001:**
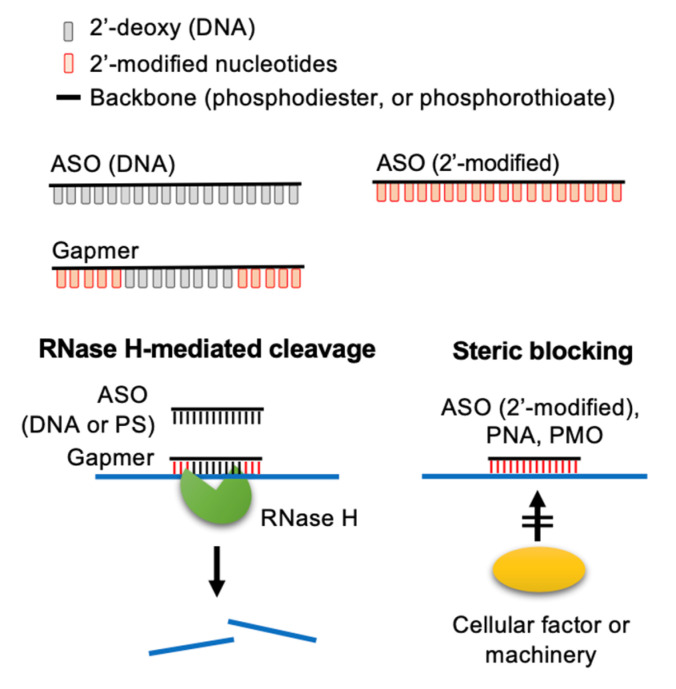
Molecular mechanisms of action of ASOs. ASOs can modulate the expression of target RNAs via two different mechanisms. Conventionally, ASOs cause RNase H-mediated cleavage of the target RNA. Additionally, 2′-O-modified ASOs and neutral DNA mimics (PMOs and PNAs) act as a steric-blocker to prevent the access of cellular factors to the target RNA. Adapted from [15], Springer Nature Limited, 2020.

**Figure 2 pharmaceutics-13-02015-f002:**
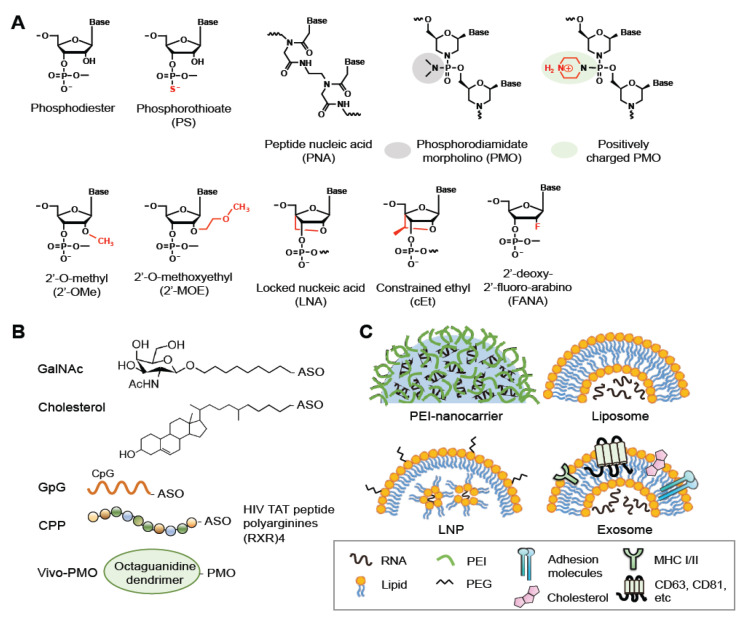
Modifications, bioconjugations and delivery vehicles of ASOs. (**A**) Structures of backbone or sugar-modified ASOs as well as PNA and PMO oligomers. (**B**) Bioconjugates of ASOs include GalNac, Cholesterol, CpG DNA and CPP (R, arginine; X, 6-aminohexanoic acid). Vivo-PMO is an PMO covalently linked to an octa-guanidine dendrimer. (**C**) Representative delivery vehicles of ASOs include EPI-nanocarrier, liposome, LNP, and exosome. Adapted from [15], Spring Nature Limited, 2020; [20], MDPI, 2021.

**Figure 3 pharmaceutics-13-02015-f003:**
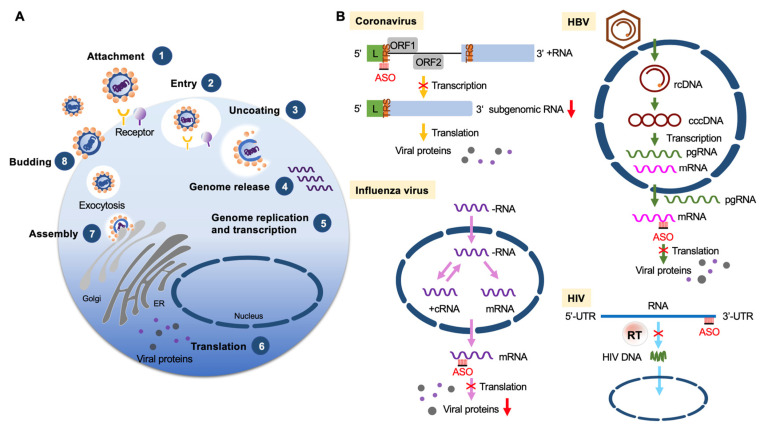
ASOs targeting viruses. (**A**) Diagram shows viral life cycle from viral attachment and entry into host cells (1, 2), genome release from the capsid (3, 4), genome replication, transcription and protein expression (5, 6), and viral assembly and release (7, 8). (**B**) ASO-based antiviral strategies. Examples are given for four different types of viruses. Coronavirus (positive-strand RNA virus): ASOs target the transcription regulatory sequence (TRS) of the RNA genome (+RNA) to reduce viral subgenomic RNA production. Influenza (negative-strand RNA virus): ASOs target viral mRNAs to reduce the production of viral nucleoprotein and matrix protein. HBV (partially double-stranded DNA virus): ASOs target a conserved sequence of viral mRNAs to reduce the translation of viral proteins. HIV (retrovirus): ASOs bind to the viral genome to interfere with reverse transcription and hence reduce viral DNA production. Abbreviations: L, leader sequence; cRNA, complementary RNA; rcDNA, relaxed circular DNA; cccDNA, covalently closed circular DNA; pgRNA, pregenomic RNA.

**Figure 4 pharmaceutics-13-02015-f004:**
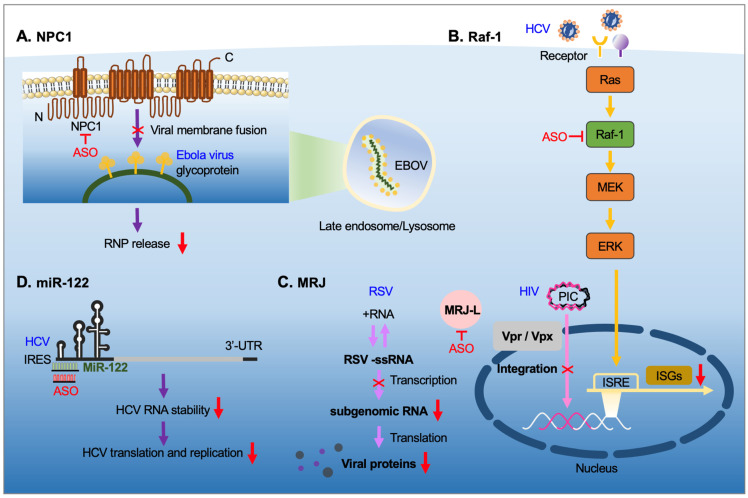
ASOs targeting host factors. Viruses take advantage of host factors for their life cycle. (**A**) NPC1 participates in membrane fusion and RNP release of Ebola virus. (**B**) Raf-1 signaling promotes HCV replication and suppresses antiviral immunity. (**C**) MRJ-L is required for the production of subgenomic RNA and mRNAs of RSV and facilitates the nuclear entry of the HIV preintegration complex. (**D**) miR-122 stabilizes the genome of HCV and promotes viral replication. Therefore, ASOs that suppress the expression of these host factors or block miRNAs can inhibit viral entry or viral genome amplification or protein production. Meanwhile, ASOs may restore antiviral activity of infected cells by suppressing the expression of certain viral or host factors. Abbreviations: ISRE, interferon-stimulated response element; ISG, interferon-stimulated gene; PIC, preintegration complex; IRES, internal ribosome entry site.

**Table 1 pharmaceutics-13-02015-t001:** Therapeutic RNAs [7,9,11].

RNA Types	Definition
ASO	Single-stranded DNA oligonucleotide with 12–25 nucleotides in length.
siRNA	Double-stranded RNA of 21–23 nucleotide in length with 2 nucleotides 3′-overhang; in general, fully complementary to the target RNA
miRNA	Double-stranded RNA of 19–25 nucleotide in length, in general partially complementary to the 3′ UTR of the target mRNA miRNA-based therapeutics include miRNA mimics and antagomirs (miRNA inhibitors).
Single-guide RNA	sgRNA is composed of a Crispr RNA (crRNA), which is a 17–20 nucleotide sequence complementary to the target DNA or RNA, and trans-activating RNA (tracr RNA), which recruits the Cas9 or Cas13 nuclease. sgRNA/Cas9 or Cas13 generates DNA or RNA cleavage.
Aptamer	A single-stranded RNA or DNA that folds into a unique structure, binding its target molecule with a high affinity
Ribozyme	An RNA molecule that has the ability to catalyze specific biochemical reactions such as RNA cleavage
Triple-forming oligonucleotide	A single-stranded oligonucleotide of 10–30 nt in length that has the potential to form triple helices with the target DNA TFO binding in general inhibits transcription or protein binding to DNA.

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
