# Peer review of "Antisense Oligonucleotide-Based Therapy of Viral Infections"

_pharmaceutics, 2021, doi:10.3390/pharmaceutics13122015_

Round 1

Reviewer 1 Report

The manuscript is centered in the revision of the use of oligonucleotides as potential antiviral compounds. The manuscript contains two parts. In the first part the authors made a general introduction on the recent advances in the field of the therapeutic use of oligonucleotides including the main mechanisms in the antisense field, the main modifications used in the antisense field, the description of the main oligonucleotide conjugates and the main formulations. In the second part the authors describe a few recent examples of oligonucleotides used as antivirals and the host factors that may influence the antiviral activity of therapeutic oligonucleotides. The first part is not bad as it is concise focusing in the main achievements of the antisense field but may be need some order and a better description of the aptamers, siRNA and microRNA field that are also very important in the development of antiviral oligonucleotides. This can be seen in the second part as a large number of the examples are examples where the authors used siRNA and microRNA mimics or microRNA inhibitors. I will say that the most interesting part of the review is the last part that includes a description of the host factors that may influence the antiviral activity. In my opinion the review can be published after a careful revision.

Comments:

Abstract. Line 3: The word “to destruct” is not appropriate. Line 7: viral infections (should be plural).

In the introduction it will be nice to rewrite the description of the nucleic-acid-based therapeutics explaining more aptamers, siRNA and microRNA fields.

Page 3. Part 2, line 3: …was reported in 1978 [17]. But ref 17 was published in 2020. I believe the authors wanted to say ref 16 that is published in 1978. Most probably they have to switch ref 16 and ref 17.

In the description of modifications of ASOs, bioconjugations of ASOs and vehicle-mediated of ASOs most of the examples are metabolic diseases (hypercholesteremia, spinal muscular atrophy, diabetes, Alzheimer disease). There are very few examples as antiviral (mainly HIV-1). May be the authors can try to add a few more examples in the antiviral field.

Figure 3 is a complex figure as the authors have tried to include the potential sites of action of ASO oligonucleotides on four different types of virus in the same figure. It is difficult to follow. Letter sizes should be bigger in some parts.

Page 8. Line 10 of the subheading 3.3 Dengue virus : ..showed that vivo-PMO targeting the terminal… is not understood.

Author Response

We thank the Reviewers for their valuable comments. Accordingly, we have modified the text and figure legends (as indicated by blue highlights) and Figure 3. We provide point-by-point responses below, and hope that the Reviewers would satisfy with this revised version.

Reviewer 1

  1. Abstract: According to the Reviewer’s suggestion, we have modified the description in the Abstract (page 1).
  2. Introduction: We have made a Table for therapeutic RNAs, in which we briefly describe each type of RNA drugs (page 3)
  3. Part 2: We have corrected reference number.
  4. Since we describe antiviral ASOs essentially in Sections 3 and 4, Section 2 thus includes some examples for non-antiviral ASOs. We indicate this point on page 5, and hope that the readers can find antiviral ASOs in Sections 3 and 4.
  5. We have modified Figure 3, and described the legends in more detail.
  6. We have modified the description (page 9).

Reviewer 2 Report

This manuscript by Tarn and co-workers is a well written and well organized review on the antisense oligonucleotide-based therapy of viral infections. The topic is of sure interests for Readers of Pharmaceutics also in light of the ongoing SARS-CoV-2 pandemic and of the search of innovative therapeutic interventions. I have only few suggestions aimed at improving the soundness of the manuscript.

  1. I would suggest to split Figure 1A and B in two separate Figures and discribe bettere in the legend (starting from the title) what is displayed in the Figure itself.
  2. In general legends to the Figures should be improved to bettere explain what is reported in each scheme. Some of them are complex and need a better explanation to result clearer for the Readers.
  3. I would add a paragraph clearly discussing the advantages and in particular the disavantages of ASOs over different RNA based therapeutic molecules, as this aspect is crucial and not well addressed by the Authors. As I mentioned, it could be a new paragraph or it could be discussed for each of the examples reported by the Authors, although I feel that the first way would be the easiest and most effective.

Author Response

We thank the Reviewers for their valuable comments. Accordingly, we have modified the text and figure legends (as indicated by blue highlights) and Figure 3. We provide point-by-point responses below, and hope that the Reviewers would satisfy with this revised version.

Reviewer 2

  1. According to the Reviewer’s suggestion, we have moved previous Figure 1B to Figure 2A.
  2. We have substantially modified the legends for all Figures (page 26).
  3. We have described both advantages and disadvantages of ASOs in the first paragraph of Discussion (page 14).